# Deep Learning-Based River Flow Forecasting with MLPs: Comparative Exploratory Analysis Applied to the Tejo and the Mondego Rivers

**DOI:** 10.3390/s25072154

**Published:** 2025-03-28

**Authors:** Gonçalo Jesus, Zahra Mardani, Elsa Alves, Anabela Oliveira

**Affiliations:** Laboratório Nacional de Engenharia Civil, Avenida do Brasil 101, 1700-066 Lisboa, Portugal; zmardani@lnec.pt (Z.M.); ealves@lnec.pt (E.A.); aoliveira@lnec.pt (A.O.)

**Keywords:** river flow forecasting, artificial intelligence, deep learning, MLP, SNIRH

## Abstract

This paper presents an innovative service for river flow forecasting and its demonstration in two dam-controlled rivers in Portugal, Tejo, and Mondego rivers, based on using Multilayer Perceptron (MLP) models to predict and forecast river flow. The main goal is to create and improve AI models that operate as remote services, providing precise and timely river flow predictions for the next 3 days. This paper examines the use of MLP architectures to predict river discharge using comprehensive hydrological data from Portugal’s National Water Resources Information System (Sistema Nacional de Informação de Recursos Hídricos, SNIRH), demonstrated for the Tejo and Mondego river basins. The methodology is described in detail, including data preparation, model training, and forecasting processes, and provides a comparative study of the MLP model’s performance in both case studies. The analysis shows that MLP models attain acceptable accuracy in short-term river flow forecasts for the selected scenarios and datasets, adeptly reflecting discharge patterns and peak occurrences. These models seek to enhance water resources management and decision-making by amalgamating modern data-driven methodologies with established hydrological and meteorological data sources, facilitating better flood mitigation and sustainable water resource planning as well as accurate boundary conditions for downstream forecast systems.

## 1. Introduction

Predicting river flow is essential for the effective and sustainable management of water resources, including timely flood emergency warnings [1]. Accurate and reliable river flow forecasts are crucial for a variety of sectors such as agriculture, water resource management, flood emergency services, and hydroelectric power production [2]. However, forecasting river flow is challenging due to the complex and dynamic nature of hydrological data, which can be influenced by precipitation, reservoir operations, land-use changes, and other factors.

Understanding river flow dynamics is vital for operational management and the safety of communities in flood-prone areas. Advanced alert systems, powered by forecasting models, enable timely evacuations and preparedness actions that help preserve lives and minimize economic losses [1]. In the context of climate change, which has led to more frequent and intense extreme weather events, accurate river flow prediction has become even more critical [3].

Furthermore, the ecological integrity of river systems is closely related to the variability of water flow. Changes in flow regimes affect sediment transport, water quality, and habitat availability for aquatic organisms. Although forecast systems for estuarine areas have improved in recent years [4,5], predicting incoming river flows remains a significant challenge [4]. Deep learning models have gained prominence in time series forecasting, including river flow prediction, due to their ability to capture complex patterns and temporal dependencies [1,5,6]. Recent comparative studies have shown that model performance can vary according to the network structure, hyperparameters, and training methodologies used [1,7,8]. Consequently, selecting and optimizing the appropriate model is essential based on the specific forecasting objectives and the available data. Sensor monitoring networks play a central role in the establishment of AI river flow models. The conceptualization and implementation of these networks is key to gathering sufficient data for reliable AI modeling. In Portugal, a very comprehensive sensor network has been deployed for 40 years, denoted as SNIRH (https://snirh.apambiente.pt/, accessed on 30 December 2024), covering multiple types of sensors and sensor variables (river flow, water level, precipitation, and dam discharges). This network provides the adequate context for developing AI models. Furthermore, the data from many sensors are available in real time, enabling the use of trained and validated models for predictive purposes. The proposed work explores the richness of the data from this sensor network and illustrates how they can contribute to improving the quality of forecasted river flows.

Given the complexities of river flow forecasting and the wide range of ML and DL approaches, we pose the following research question:

“*Can the systematic deployment of MLP models—optimized through extensive hyperparameter tuning and evaluated in comparison with alternative machine learning(ML) techniques—provide reliable and robust short-term river flow forecasts in dam-regulated rivers, under both normal and raised flow conditions?*”

Our hypothesis is that MLP-based forecasting, when carefully compared and optimized alongside models such as LSTM, SVM, ELM, and RF, will generally yield robust and reliable river flow predictions. We expect that MLPs will demonstrate competitive performance under normal flow conditions and produce reasonably accurate results in more challenging high-flow scenarios. Although we do not claim that MLPs will always outperform all alternatives in extreme events, we anticipate that they offer a practical balance between forecasting ability and computational efficiency.

In this paper, we focus on applying MLPs for river flow forecasting and evaluate their efficacy relative to other ML and DL methods. The models considered include MLPs, LSTMs, SVMs, RFs, and ELMs. This initial exploration provides insights into the strengths and limitations of each approach and supports our decision to focus on MLPs for an in-depth analysis and for establishing forecast services that integrate with estuarine predictions.

Based on our findings, we define two specific scenarios for the Tejo River and one for the Mondego River, targeting the prediction of river discharge for the next three days using MLPs. These models integrate data from downstream dams, including river discharge, reservoir outflows, reservoir levels, and precipitation, to improve the prediction precision. Our objective is to demonstrate the applicability and robustness of MLPs in diverse hydrological and hydraulic contexts.

The research aims to develop and refine AI models that function as remote services, providing river flow forecasts for the current day and the following two days. These models combine advanced data techniques with reliable hydrological and meteorological data to improve water management decisions.

This study is part of the CONNECT project, within the Coastal Downscaling Program of the Copernicus Marine Service, and the ATTRACT project (Digital Innovation Hub for Artificial Intelligence and High-Performance Computing). Both projects focus on developing and implementing AI-based computational services to facilitate industry uptake of AI-driven products. The remainder of the paper is organized as follows. Section 2 reviews the relevant literature and previous studies. Section 3 outlines the case studies and methodology. Section 4 provides a concise formulation of the fundamental concepts and mathematical foundations of the ML and DL models employed in this study, including MLP, LSTM, SVM, ELM, and RF. Section 5 presents the application of the methodology and compares the performance of different models. Finally, Section 6 discusses the main findings and suggests directions for future research.

## 2. Related Work

River flow forecasting has received significant attention in the last decades due to its critical role in water resource management, flood mitigation, and environmental conservation [9]. The complexity of hydrological systems requires the development and application of advanced predictive models. ML and deep learning (DL) models have emerged as prominent tools in this domain, offering enhanced capabilities to capture the intricate patterns inherent in river flow data [10]. However, each method has its own set of advantages and limitations, and a comprehensive understanding of them is essential to advance the field.

### 2.1. ML and DL Models for River Flow Forecasting

A diverse array of ML and DL models have been used for river flow forecasting, each with unique strengths and applications. Below, we briefly review these methods and critically analyze their advantages and disadvantages for time series forecasting:Autoregressive Integrated Moving Average (ARIMA)ARIMA models are traditional time series forecasting tools that capture temporal trends, seasonality, and time-dependent patterns in data. They are particularly effective in analyzing historical river flow data and providing baseline forecasts [11,12]. Despite their interpretability, ARIMA models often struggle with non-stationary data and the nonlinear patterns typical of hydrological processes [9].Linear Regression (LR)LR models the relationship between river flow and external factors such as precipitation and temperature through a linear equation. Although simple and interpretable, LR is limited in its ability to capture nonlinear interactions and complex dependencies that often characterize river flow processes [13].Multilayer Perceptron (MLP)MLPs are feedforward neural networks capable of capturing nonlinear relationships in data. They are well suited for short-term river flow predictions, especially when high temporal resolution data are available. While computationally efficient and flexible, MLPs require extensive hyperparameter tuning, and their performance is sensitive to the quality of input data. Furthermore, the default optimization algorithm (gradient descent) commonly used in MLPs often converges to local minima, particularly when handling highly stochastic time series data like river streamflow. This limitation, along with the risk of overfitting, can lead to inaccurate predictions, thus necessitating more advanced optimization strategies [14].Support Vector Machines (SVMs)SVMs are supervised learning models that predict continuous outcomes by finding the optimal hyperplane that separates data. They effectively model complex nonlinear relationships using kernel functions. In hydrological applications like river flow forecasting, this sensitivity may result in unreliable forecasts when the data display significant variations or noise, thereby decreasing overall forecasting accuracy [15].Random Forests (RFs)RFs are ensemble learning methods that construct multiple decision trees and merge their results to improve predictive accuracy and control overfitting [16]. As noted in [15], RF-based approaches sometimes underestimate extreme values in longer forecast horizons (e.g., 2- and 3-h projections), thereby diminishing their ability to accurately predict severe hydrological phenomena.Extreme Learning Machines (ELMs)ELMs are single-hidden-layer feedforward neural networks with randomly assigned weights and biases, offering rapid training speeds and good generalization performance. Sometimes, ELMs face accuracy limitations when modeling complex and highly dynamic hydrological scenarios, despite their efficiency. Additional disadvantages include their dependence on random initialization, which can lead to inconsistent performance across experiments, and the need for a large number of hidden neurons to achieve competitive accuracy, which increases both computational complexity and the risk of overfitting [17].Recurrent Neural Networks (RNNs)RNNs are designed to handle sequential data by maintaining a memory of previous inputs, making them effective for modeling dynamic systems such as river flow. However, standard RNNs often face challenges such as vanishing gradients that hinder their ability to learn long-term dependencies. Furthermore, they can suffer from instability issues such as gradient explosion, require the fine-tuning of learning rates and other meta-parameters, and generally struggle to capture long-term dynamics in complex hydrological time series [18].Long Short-Term Memory (LSTM)LSTMs, an advanced RNN architecture, overcome the vanishing gradient problem and capture long-term dependencies in time series data [19]. However, the complexity and computational requirements of LSTMs, as highlighted by Rahimzad et al. (2021), can become significant challenges, especially with large datasets [20].Convolutional Neural Networks (CNNs)Originally developed for spatial data, CNNs have been adapted for time series forecasting by capturing local patterns in hydrological data, such as precipitation and topography. Because CNNs are inherently designed to extract spatial features, they can struggle with modeling long-term temporal dependencies in hydrological processes and may require large amounts of training data and careful kernel size design to effectively capture the sequential nature of the data [21].Gated Recurrent Unit (GRU)GRUs are streamlined versions of LSTMs with fewer parameters, resulting in faster training while still effectively capturing temporal relationships. They may not capture the fine temporal dynamics as well as LSTMs in all cases, but they offer a good compromise between complexity and performance. Furthermore, GRUs may sometimes fall short when modeling very complex or long-term dependencies in hydrological time series compared to more sophisticated architectures [22].Positive and Negative Perceptron (PNP)PNPs incorporate both positive and negative contributions within their architecture, aiming to capture diverse hydrological characteristics more effectively. Being relatively new, further research is needed to establish their stability and reliability in river flow prediction. Furthermore, their innovative structure may require more complex tuning and extensive validation to ensure robustness under different hydrological conditions, and their relative performance against traditional models remains to be comprehensively evaluated [23].Attention-Based Neural Networks (AttNet)Recent studies have shown that attention-based models can significantly enhance streamflow forecasting by focusing on the most relevant temporal features [24,25]. Despite promising results, the increased complexity and computational demands of these models can be challenging in operational settings. Moreover, as highlighted by Liu et al. (2024) and Lee et al. (2024), these models often require extensive data segmentation, hyperparameter tuning, and significant computational resources to effectively manage and interpret complex hydrological data, which can impede their real-time application.Hybrid ModelsHybrid models integrate multiple methodologies—such as combining wavelet transforms with ML algorithms—to capture both linear and nonlinear patterns in river flow data. Hybrid models, by combining multiple methods such as wavelet transform and ML algorithms, demonstrate more advanced forecasting capabilities by exploiting both linear and nonlinear components of hydrological datasets [26,27].

The recent literature emphasizes that a deep understanding of the trade-offs between these methods is crucial. Comprehensive evaluations that critically analyze the strengths, weaknesses, and suitability of these models for specific forecasting tasks help inform the selection of the most appropriate model for a given hydrological challenge [28,29].

### 2.2. Research Overview

In our study, we identify a select group of ML and DL models that are particularly well suited to our case studies in river flow forecasting. This group includes LSTM networks, ELMs, RFs, SVMs, and MLPs. LSTM networks have emerged as a dominant approach in hydrological modeling due to their ability to effectively capture temporal dependencies and nonlinear patterns in time series data. For example, Rahimzad et al. [20] found that LSTM networks consistently outperformed LR, MLP, and SVM in river flow prediction. Building on this, Bakhshi Ostadkalayeh et al. [30] enhanced LSTM models by integrating Kalman filtering to significantly reduce prediction errors. Similarly, Ho et al. [31] applied multi-step-ahead LSTM models to improve sluice gate operations in Vietnam, while Cho and Kim [32] merged LSTM with the WRF-Hydro model for improved streamflow predictions. Additional studies by Xie et al. [33], Xiang and Demir [34], and Ni et al. [27] further highlighted the potential of hybrid LSTM-based approaches in capturing seasonal variations and extreme hydrological events. Nguyen et al. [35] developed a deep neural network with LSTM layers to predict flow in the Mekong River Basin. Their results showed that the model could accurately capture seasonal patterns and abrupt changes in flow. In summary, the LSTM components were key to modeling the complex temporal dependencies in the Mekong flow regime. The results indicated that, compared with traditional approaches, the LSTM-based model achieved robust and reliable predictions under different hydrological conditions, demonstrating its strong potential for large-scale river basin applications. In addition, Hunt et al. [36] demonstrated that integrating LSTM networks into streamflow forecasting frameworks can significantly enhance the accuracy of predictions over the Western United States.

ELMs have gained traction in hydrological forecasting for their rapid training capabilities and efficiency in handling nonlinear relationships [17]. Furthermore, Bărbulescu and Liu [37] compared various AI methods for river water discharge forecasting and found that, although advanced DL approaches often provide high accuracy, ELMs offer a competitive alternative due to their significantly faster training times and lower computational demands, making them suitable for real-time forecasting applications. RFs, introduced by Breiman [16], have also been shown to outperform other algorithms in terms of accuracy and robustness under varying hydrological conditions [38]. Additionally, Islam et al. [39] investigated the use of Random Forest regression with remotely sensed data to predict streamflow in a snowmelt-dominated watershed. Their study showed that a customized RF approach outperformed the physically based SWAT model—especially when trained over long periods—with snow depth and minimum temperature being the most critical predictors. SVMs have been widely used in hydrological modeling because of their proficiency in capturing complex nonlinear relationships, although they can be computationally intensive for large-scale applications [15]. Notably, Mahmood et al. [40] investigated reservoir inflow forecasting for the Haditha Reservoir in Western Iraq using SVM. Their study showed that SVM can be effectively applied in data-poor conditions and achieved competitive performance compared to neural network models, emphasizing that SVM remains a viable option for flood and inflow forecasting in complex hydrological settings.

Dibike and Solomatine [41] demonstrated the effectiveness of ANNs, especially MLPs, in the capture of complex hydrological patterns. Their study compared MLPs with a conceptual rainfall–runoff model and found that the ANN approach performed slightly better than traditional methods in predicting river flow. Brandão et al. [42] investigated flood forecasting in an ungauged basin using the Paranaíba River as a case study. This work demonstrates that artificial neural networks, specifically MLPs, can effectively capture the nonlinear dynamics of flood events in data-poor environments, offering a promising tool for early flood warnings.

Finally, MLPs remain a fundamental component in river flow forecasting due to their flexibility and capacity to model nonlinear patterns, especially when combined with advanced feature engineering techniques like Principal Component Analysis (PCA) [14].

## 3. Methodology

This section describes the case studies and the methodology for forecasting river flow (Figure 1) for the next one, two, and three days. The workflow comprises three main steps: preprocessing, model development, and model validation and forecasting. Each step is tailored to the data available for each basin, with the selection of data stations being based on expert knowledge of river dynamics and basin-specific characteristics.

### 3.1. Case Studies: Tejo and Mondego Rivers

We focus on forecasting the daily river flow for two important Portuguese rivers: the Tejo (Tagus) and the Mondego. Data for both rivers were sourced from Portugal’s national water resource information system, the Sistema Nacional de Informação de Recursos Hídricos (SNIRH) [43], which provides comprehensive hydrological and meteorological data essential for accurate forecasting models.

#### 3.1.1. Tejo River

The Tagus basin (Figure 2) is one of the largest in the Iberian Peninsula. The Tejo River spans 1007 km and covers a basin area of 80,626 km^2^ (24,845 km^2^ in Portugal). Its flow regime is primarily controlled by several dams located on both the main river and its tributaries [44].

Flood events in the Tejo River are frequent, with the 1979 floods being particularly severe. Reliable river flow predictors are therefore essential. Accurate forecasts are also important for the Tagus estuary (developed in the CONNECT project), where the freshwater/saltwater balance critically influences the water quality [45].

For the Tejo River, two datasets based on daily discharges and precipitation were used to forecast the daily river flow at Almourol hydrometric station for the period from 1 October 1984 to 26 September 2023.

#### 3.1.2. Mondego River

The Mondego River basin (Figure 3) covers 6645 km^2^, making it the second largest basin entirely within Portugal. Stretching 234 km, the Mondego River originates in mountainous regions, flows into a wide alluvial flood plain, and ultimately empties into the Atlantic Ocean. Its flow is regulated by dams—Aguieira, Raiva, Fronhas, and Açude-Ponte Coimbra—constructed during the 1980s.

Flooding is a major concern in the Mondego basin [46]. Infrastructure such as lateral dikes and flood-control structures (e.g., at the Açude-Ponte in Coimbra) have been constructed to mitigate flood impacts. Accurate river flow predictions are critical for both flood management and for estimating freshwater intake for the Mondego estuary forecast, where freshwater flows significantly affect the salt balance during heavy precipitation events.

For the Mondego River, only stations with long daily records and real-time data were selected (excluding precipitation data). The dataset consists of daily discharges from the Aguieira, Raiva, Fronhas, and Açude-Ponte Coimbra dams, covering the period from 4 November 1997 to 9 March 2024. The AI model was developed to predict daily discharges at the Açude Ponte Coimbra (12G/01AE) station.

### 3.2. AI Model Construction

The process of developing robust *artificial intelligence (AI)* river flow forecasting models consists of multiple essential stages, each specifically tailored to guarantee the precision and dependability of the predictions. This section offers a comprehensive explanation of the steps required, beginning with data preprocessing, then moving on to model training, and concluding with the forecasting phase.

#### 3.2.1. Data Collection

Data for this investigation were obtained from national hydrometric and meteorological monitoring networks to guarantee extensive coverage and dependability. The datasets included hydrological variables, such as river discharge and effluent flow, and meteorological data, such as precipitation and temperature, pertinent to river flow dynamics. Two scenarios were examined: one concentrated on hydrological data from dam-regulated discharges, while the other included meteorological inputs from nearby regions. These scenarios were created to investigate the prediction capacity of AI algorithms to display river flows in certain target areas.

#### 3.2.2. Preprocessing Steps

The following preprocessing steps were taken to ensure data quality, consistency, and integrity:Data Synchronization: Initially, datasets from various stations, including hydrometric and meteorological stations monitored by the Sistema Nacional de Informação de Recursos Hídricos (SNIRH), are acquired and formatted into a consistent structure. This involves parsing date–time information, normalizing measurement units, and synchronizing records from different stations to establish a unified timeline. Normally, the dataset is loaded from a comma-separated values (CSV) file, downloaded from the SNIRH website, and the date column is converted to a date–time format. The date column is then set as the index of the DataFrame to facilitate time series analysis.Missing Data Handling: Ensuring the completeness of the data is essential to preserve the accuracy and reliability of the dataset. To properly handle missing values, several techniques are employed, including linear interpolation and forward and backward filling. However, due to the nature of the river flow data, filling in missing values can introduce inaccuracies. Therefore, we create a set of functions to promote these assessments.Feature Selection: It is crucial to identify and select the key variables that significantly influence the predictive model’s performance. These variables include historical river flow discharge measurements, meteorological data (such as precipitation), and other relevant factors, such as dam discharge rates. For this study, we promote several selections based on the performance of the constructed model.Temporal Resampling: To achieve consistency, the data’s resolution is standardized through temporal resampling. The dataset is resampled to a daily frequency to ensure uniform time intervals. This step aggregates the data and fills any missing dates with interpolated values.Alignment of Common Periods: It is crucial to synchronize datasets from several stations so they overlap within the same time periods. This guarantees that models are trained on datasets encompassing all pertinent characteristics within the same time frame. Due to the presence of significant missing values, we focus on combining periods with minimal missing data to ensure the integrity of the dataset.The get_common_periods_sections function is used to identify periods with a maximum of 10 missing values. This approach helps maintain the validity of the river flow data while ensuring enough data are available for model training. The steps are shown in Algorithm 1, which includes combining datasets, figuring out missing values, and selecting acceptable time segments based on certain criteria.



**Algorithm 1** Identify common periods with minimal missing values and fill missing data.**Require:**    –  dataframes: List of DataFrames from different stations    –  max_missing: Maximum allowed missing values per day (e.g., 10)    –  min_required_period: Minimum number of consecutive days required**Ensure:**    –  common_periods: List of start and end dates with minimal missing data    –  filled_data: DataFrames with missing values filled by interpolation 1: Initialize common_periods as an empty list 2: Initialize filled_data as an empty list 3: Merge all dataframes on the date–time index using an outer join 4: Calculate the total number of missing values per day 5: Create a Boolean mask where missing values ≤ max_missing 6: Find continuous True segments in the mask 7: **for** each continuous segment **do** 8:     **if** length of segment ≥ min_required_period **then** 9:         Append (start_date, end_date) to common_periods 10:       Extract data for the segment 11:       Fill missing values in the segment using interpolation 12:       Append filled segment to filled_data 13:     **end if** 14: **end for** 15: **return**
common_periods, filled_data



Transformation to Supervised Learning Format: Time series data are converted into a format suitable for supervised learning. The series_to_supervised function transforms the time series data into a supervised learning problem by creating lagged versions of the input features. This transformation allows the model to learn temporal dependencies in the data. The function creates input sequences of length $n_{in}$ and output sequences of length $n_{out}$. To generate input–output pairings, Algorithm 2 methodically shifts the data. The future values ($t+1,\ldots ,t+n_{out}$) function as targets, and the lag features ($t-n_{in},\ldots ,t$) as model inputs.


**Algorithm 2** Transform time series to supervised learning format.**Require:**    –  data: Time series data as a DataFrame    –  n_in: Number of lag observations as input (e.g., 3)    –  n_out: Number of observations as output (e.g., 1)**Ensure:**> supervised_data: Transformed DataFrame suitable for supervised learning 1: Initialize cols as an empty list 2: **for** i = −n_in 0 **do** 3:   cols.append(data.shift(i)) 4: **end for** 5: **for** j = 1 n_out **do** 6:   cols.append(data.shift(−j)) 7: **end for** 8: Concatenate cols along the column axis 9: Drop all rows with NaN values 10: Rename columns appropriately (e.g., var(t − n), …, var(t), var(t + 1), …) 11: **return** supervised_data

#### 3.2.3. Model Development

The model training stage involves splitting the data into training and testing sets, defining the model architecture, and training the model. The key steps are as follows:Data Partitioning: The combined supervised data are split into training and testing sets using an 80–20 ratio. The training set is used to train the model, while the testing set is used to evaluate its performance.Model Architecture Definition: The structure of the ML models is determined according to the particular needs of the forecasting task. This involves choosing the appropriate number of layers, neurons per layer, activation functions, and other architectural characteristics for models such as *LSTM*, *MLPs*, *ELMs*, *RFs*, and *SVMs*. We mainly used the Keras library to define the model. Multiple instances of these models were trained and selected based on performance. For example, for MLPs, we used multiple dense layers with ReLU activation functions and dropout layers to avoid overfitting. The input dimension of the initial layer was set to the number of features in the training data, while the output dimension was set to the number of forecasting steps (three days).Hyperparameter Optimization: To improve model performance, hyperparameters were fine-tuned using grid search. The goal of this stage was to determine the best combination of hyperparameters that maximizes the predicted accuracy of the model. Depending on the model configuration, we varied the number of neurons per layer, the number of epochs, L2 regularization values, dropout rates, batch sizes, optimizers, and early stopping patience. Although comprehensive hyperparameter tuning improves model accuracy for a specific dataset, it creates issues with scalability and adaptability. While grid search enhances model performance under a set of conditions, it may fail to consider seasonal fluctuations, climate change, or human activities over time. Other tuning methodologies can be pursued, such as Bayesian Optimization or meta-learning, to counteract the diminishing model efficacy if hyperparameters are not adjusted.Model Training: Our models are trained using the Adam optimizer and mean squared error (MSE) as the loss function. Early stopping is used to monitor the validation loss and prevent overfitting. The model is trained for a specified number of epochs, and the best model weights are saved based on the validation loss.

#### 3.2.4. Model Validation and Forecasting

The forecasting stage uses the trained models to generate predictions for three days. The key steps are as follows:Prediction Generation on Test Data: The trained models generate predictions based on the test data. To evaluate model performance, the root mean squared error (RMSE) is calculated for each forecasting step (today, tomorrow, and the day after tomorrow).Validation with Future Data: The models are validated on a separate validation set containing recent data. For example, we currently use the entire 2023 dataset. The validation data are preprocessed and transformed in the same manner as the training data. Each model generates predictions for the validation period, and the RMSE is calculated for each forecasting step.

Model validation ensures predicted accuracy during testing; nevertheless, it does not assure long-term durability amongst changing hydrological regimes. To prevent such scenarios, regular retraining with recent data or methods such as transfer learning or gradual learning can help preserve the model accuracy.

## 4. Theoretical Background

The fundamental concepts and essential equations for the *ML* and *DL* models used in this study are explained in this section.

### 4.1. Multilayer Perceptron

A multilayer perceptron [47,48] is a feedforward neural network composed of an input layer, one or more hidden layers, and an output layer. Each neuron in a hidden layer computes(1)$$h=f\left(Wx+b\right),$$
where x is the input vector, W is the weight matrix, b is the bias vector, and f(·) is a nonlinear activation function (e.g., ReLU, sigmoid, or tanh). The output is obtained by applying a similar operation at the final layer.

### 4.2. Long Short-Term Memory Networks

A Long Short-Term Memory network [49] is a specialized type of Recurrent Neural Network (RNN) designed to capture long-term temporal dependencies in sequential data. The internal gating mechanisms in an LSTM cell are defined as(2)$$\begin{array}{cccc} i_{t} & =\sigma \left(W_{i}\left[h_{t-1},x_{t}\right]+b_{i}\right),\; & f_{t} & =\sigma \left(W_{f}\left[h_{t-1},x_{t}\right]+b_{f}\right), \end{array}$$(3)$$\begin{array}{cccc} o_{t} & =\sigma \left(W_{o}\left[h_{t-1},x_{t}\right]+b_{o}\right),\; & {\tilde{c}}_{t} & =\tanh\left(W_{c}\left[h_{t-1},x_{t}\right]+b_{c}\right), \end{array}$$(4)$$\begin{array}{cccc} c_{t} & =f_{t}⊙c_{t-1}+i_{t}⊙{\tilde{c}}_{t},\; & h_{t} & =o_{t}⊙\tanh\left(c_{t}\right), \end{array}$$
where $i_{t},f_{t},o_{t}$ denote the input, forget, and output gates, respectively; $c_{t}$ is the cell state; $h_{t}$ is the hidden state; and ⊙ is element-wise multiplication. The function σ(·) is the sigmoid activation, and tanh(·) is the hyperbolic tangent.

### 4.3. Support Vector Machine

Support vector regression [50] aims to find a function f(x) that deviates from the actual target values by at most ϵ. This is formulated as(5)$$\begin{array}{cc} \min_{w,b,\xi ,\xi^{*}}\; & \frac{1}{2}{∥w∥}^{2}\;+\;C\sum_{i=1}^{N}\left(\xi_{i}+\xi_{i}^{*}\right) \end{array}$$(6)$$\begin{array}{cc} \mathrm{subject}\;\mathrm{to}\;\; & \left\{\begin{array}{c} y_{i}-\left(w^{⊤}x_{i}+b\right)\;\leq \;\epsilon +\xi_{i},\\ \left(w^{⊤}x_{i}+b\right)-y_{i}\;\leq \;\epsilon +\xi_{i}^{*},\\ \xi_{i},\;\xi_{i}^{*}\;\geq \;0, \end{array}\right. \end{array}$$
where *C* is a regularization parameter, and $\xi_{i},\;\xi_{i}^{*}$ are slack variables controlling the allowed error.

### 4.4. Extreme Learning Machine

An extreme learning machine [17] is a single-hidden-layer feedforward network. The hidden layer weights and biases are randomly initialized and remain fixed, while the output weights are computed analytically via(7)$$\beta =H^{†}\;T,$$
where H is the hidden layer output matrix, T is the target matrix, and $H^{†}$ is the Moore–Penrose pseudoinverse of H.

### 4.5. Random Forest

A Random Forest [16] is an ensemble of decision trees. For a given input x, each tree *t* outputs a prediction $h_{t}\left(x\right)$. The final prediction is the average of all tree outputs:(8)$$\hat{y}\left(x\right)=\frac{1}{T}\sum_{t=1}^{T}h_{t}\left(x\right),$$
where *T* is the total number of trees.

## 5. Application to the Case Studies

We evaluated the performance of various ML models, including LSTM networks, MLP, SVM, ELM, and RF. Through systematic hyperparameter tuning and performance comparison, we identified the most effective model for river flow forecasting in each case study.

### 5.1. Comparison of Models Performance for Tejo River and Selection of MLP

The comparative performance of different ML and DL models in forecasting daily river discharge for the Tejo River.

The evaluation focuses on the RMSE (in m^3^/s) for predictions for today and tomorrow.

To perform a thorough analysis, we designed several experiments using different input stations within the Tejo River, with the Almourol station as the target for discharge predictions. The selected input stations include the following:0. Castelo de Bode Average daily dam outflow discharge (m^3^/s)1. Castelo de Bode Reservoir (m)2. Fratel Average daily dam outflow discharge (m_3_/s)3. Fratel Reservoir water level (m)4. Almourol Daily Average discharge (m_3_/s)

For the evaluation of different models, we categorized the experiments into four scenarios built with different input variables:Scenarios **a**, **b**, and **c**: Validation period from 2022-08-07 to 2023-09-04.Scenario **d**: Validation period from 2003-03-31 to 2004-11-07.

The scenarios were chosen based on hydrological significance, data accessibility, forecasting goals, and model complexity, with a focus on critical hydrometric stations that have minimal data gaps and extensive datasets. We also considered scenarios with extreme river flow conditions (flood events) to assess model robustness under both typical and exceptional circumstances. First, we preprocessed all time series without missing values and combined them into a comprehensive dataset. We then trained MLP and LSTM models to evaluate their ability to capture temporal relationships. Based on their performance, the MLP was selected as the preferred model. Subsequently, the MLP was compared with other ML models (i.e., ELM, SVM, and RF). While ELM and SVM did not achieve adequate prediction performance, RF demonstrated strong accuracy for 1-day forecasts but poorer performance for 2-day predictions. Overall, the MLP consistently achieved a lower RMSE across most scenarios and forecasting horizons using systematic hyperparameter tuning (including grid search).

Table 1, Table 2, Table 3, Table 4 and Table 5 show the configurations of the LSTM, MLP, ELM, SVM, and RF models for the different scenarios. The abbreviations used in the model configuration tables are explained in Appendix A. The comparison of models performance is presented in Table 6, and it is possible to extract the following key insights:**Scenario a** (Input: [0,2] → [4], 2-day forecast):–The MLP configurations yielded RMSE values of 162.65 (today) and 227.02 (tomorrow) in one configuration, while LSTM models produced slightly higher errors (168.70 and 239.71, respectively).–SVM reported RMSE values of 367.23 and 367.31 in one configuration and 203.01 and 231.81 in another, indicating sensitivity to hyperparameter selection.–Although RF achieved an RMSE of 155.71 for the 1-day forecast, its error increased to 264.51 for the 2-day prediction.–ELM results were generally higher, with one configuration reporting 326.42 and 334.82, and another with 307.51 and 308.27.**Scenario b** (Input: [0,2,3] → [4], 2-day forecast):–The MLP produced RMSE values of 152.71 and 213.88 in one configuration and 149.14 and 227.87 in another, suggesting that including an additional input (from station 3) improved performance.–The LSTM model yielded RMSE values of 141.85 and 216.48, while SVM and RF achieved 198.12, 230.45 and 148.79, 261.46, respectively.–ELM reported values of 218.08 and 255.18.**Scenario c** (Input: [0,1,2,3,4] → [4], 2-day forecast):–The MLP achieved RMSE values as low as 136.71 and 206.49 in one configuration and 141.39 and 217.94 in another.–LSTM values were 136.02 and 212.92, while SVM produced 153.17 and 200.09.–RF and ELM in this scenario recorded RMSE values of 149.53, 257.72 and 103.82, 179.47 (with the latter configuration for ELM highlighting the potential for lower error in one output), respectively.**Scenario d** (Input: [0,2,4] → [4], 1-day forecast):–The MLP reported an RMSE of 104.53 for the 1-day forecast, which is slightly better than that of SVM, at 105.75, and notably lower than those of LSTM, at 121.51, and RF, at 119.11.–ELM, however, showed a higher RMSE of 223.36 (with one additional configuration at 356.93 in the LSTM column).

These results indicate that each model’s performance is closely tied to its hyperparameter configuration (Table 1, Table 2, Table 3, Table 4 and Table 5) and the specific input combination used. In summary, the comparative analysis shows that, although several models can yield competitive results in specific settings, the performance of the MLP is promising for our dataset. This evaluation supports further investigation of the MLP approach for river flow forecasting in the Tejo basin, while acknowledging that model performance can vary with different configurations, training periods, and input selections.

For example, in one experimental configuration (scenario a) for a two-day forecast, the training dataset consists of 8049 samples with an input shape of (8049,40) and an output shape of (8049,2), while the validation dataset has 373 samples with shapes of (373,40) and (373,2). The RMSE values observed in this configuration are 162.65 for the first day and 227.02 for the second day. By contrast, another configuration (scenario d) flattened for the MLP, resulting in a training set of 1552 samples with shape (1552,60), a testing set of 389 samples with shape (389,60), and a validation set of 547 samples with shape (547,60). This configuration, designed for a one-day forecast, achieves an RMSE of 104.53. The differences in RMSE can be attributed to several factors:Forecast Horizon: A two-day forecast inherently introduces more uncertainty, leading to higher RMSE values than a one-day forecast.Validation Period and Training Duration: The experimental configurations are validated over different periods. A validation period that captures higher discharge variability or extreme events results in higher RMSE values compared to a period with more stable flows.Input Features and Model Configuration: Although both configurations are based on dam data, variations in the number of input features (40 in scenario a vs. 20 in scenario d, with subsequent flattening to 60) and differences in the hyperparameter settings impact the model’s ability to capture the underlying hydrological dynamics.

Thus, the observed RMSE values, 162.65 and 227.02 in one configuration versus 104.53 in another, reflect the influence of different forecast horizons, validation periods, and input configurations on model performance.

The MLP model regularly exceeds other ML models in river flow forecasting as evidenced by its reduced RMSE values in most cases. This success is primarily due to its capacity to precisely model complex nonlinear relationships in the data, which is especially advantageous for forecasting sophisticated river flow dynamics. Moreover, MLP provides considerable computational efficiency relative to deep learning models such as LSTM (Long Short-Term Memory), requiring fewer computational resources while maintaining strong performance. This makes MLP a viable option for handling large datasets or when computing resources are limited. The adaptability and scalability of MLP architectures allow customization to suit various types and sizes of datasets, thus enhancing their relevance to diverse river flow forecasting applications [14,51]. Furthermore, previous studies have supported the effectiveness of MLP in forecasting time series, affirming its reliability in projecting river flow based on historical data [10,52,53]. These combined advantages position MLP as our study’s leading model for river flow prediction, outperforming alternatives such as SVM, RF, and ELM.

Hyperparameter Optimization: MLP requires the fine-tuning of parameters such as the number of neurons in hidden layers, learning rate, and regularization coefficients. Identifying the optimal configuration can be time-consuming and necessitates extensive experimentation, particularly with large and complex datasets [28,54].Overfitting: MLP is susceptible to overfitting when the model complexity exceeds the available training data. Overfitting can lead to excellent performance on the training dataset but poor generalization to unseen data. Although regularization techniques such as L2 regularization and dropout can mitigate this issue, they require meticulous calibration to balance model complexity and performance [55].

Despite these limitations, the MLP model remains effective in predicting daily river discharge within the scope of this investigation. The dataset is relatively stable, and the primary challenge lies in identifying complex patterns in the time series data.

MLP is selected for its effectiveness in handling time series data. The architecture consists of an input layer, two hidden layers, and an output layer designed to predict river discharge for three consecutive days: today, tomorrow, and the day after tomorrow. Hyperparameters are fine-tuned using grid search to optimize performance, and early stopping is implemented to prevent overfitting during training.

### 5.2. Model Configurations and Forecasting Results

Here, after choosing the MLP model, we narrow our focus to two specific scenarios for the Tejo River each with different input features and configurations and one for the Mondego River. For the Tejo River, Scenario 1 and Scenario 2 are designed to explore how distinct sets of features and hyperparameter settings affect forecasting performance. For the Mondego River, we evaluate a single scenario.

Figure 4 and Figure 5 illustrate the common data periods for Tejo River scenarios 1 and 2, respectively, while Figure 6 shows the data period for the Mondego River scenario.

This section outlines the configurations for each scenario and presents the corresponding forecasting results, including key performance metrics such as RMSE across various forecast horizons.

Figure 4 and  Figure 5 display the common data periods at the measuring stations for the Tejo River, while Figure 6 shows those for the Mondego River.

Selected features: Table 7 lists the variables chosen for the Tejo River configurations, and Table 8 provides the selected variables for the Mondego River.

For each scenario, the dataset was divided into periods for training, testing, and validation. Table 9 summarizes the size of the dataset for each phase.

These tables and figures together ensure that the selection of variables, data coverage, and dataset splits are clearly documented, thereby supporting the subsequent analysis of model configurations and forecasting results.

#### 5.2.1. Mondego River

For the scenario, three distinct MLP models are developed with varying hyperparameters. Their performance in terms of RMSE (in m^3^/s) across different forecasting horizons is presented in Table 10.

The RMSE values for each configuration across the 3-day forecast horizons are presented in Table 11.

Based on these results, MLP1 is identified as the best-performing model due to its consistent and lower RMSE values across the "Tomorrow" and "Day After" forecasts. Consequently, MLP1 is selected for validation and further application in Scenario (1), which spans from 1 January 2019 to 1 May 2020.

These definitions and the accompanying parameter summary clarify the detailed hyperparameters for each MLP configuration and provide valuable insights into how the architectural choices and training parameters affect the model’s ability to capture the nonlinear dynamics of river flow.

#### 5.2.2. Tejo River

For the Tejo River, we evaluated two distinct configurations over the validation period from 7 August 2023 to 13 August 2024.

Scenario (1): Two hidden layers with 90 neurons each, trained for 100 epochs.Scenario (2): Two hidden layers with 150 neurons in the first hidden layer and 40 neurons in the second hidden layer, trained for 300 epochs.

These configurations ensure that custom models are applied to each dataset, effectively capturing the unique influences of river flow, dam operations, and precipitation on discharge predictions.

### 5.3. Performance Metrics

To evaluate the accuracy of the forecasts, we use the RMSE and bias, defined as follows (see [56]):(9)$$\mathrm{RMSE}=\sqrt{\frac{1}{n}\sum_{i=1}^{n}{\left(P_{i}-O_{i}\right)}^{2}},$$(10)$$\mathrm{Bias}=\frac{1}{n}\sum_{i=1}^{n}\left(P_{i}-O_{i}\right),$$
where $P_{i}$ is the predicted value, $O_{i}$ is the observed value, and *n* is the total number of observations.

RMSE (Equation (Equation 9)) quantifies the overall magnitude of prediction errors by squaring individual differences before averaging, thus placing a heavier penalty on larger errors. A lower RMSE indicates better agreement between predictions and observations.Bias (Equation (Equation 10)) measures the systematic offset between the model and the observations. A positive bias means the model tends to overpredict, while a negative bias indicates underprediction.

### 5.4. Model Evaluation

The performance of the MLP models was assessed by comparing their predictions with the observed river flow discharge values for both rivers in different scenarios. A smaller RMSE and a bias closer to zero indicate better forecasting ability. Table 12 presents the RMSE and bias values for each scenario and forecast horizon.

Figure 7 and  Figure 8 present a comparison between the measured and predicted values for the Tejo River in Scenario (1). Similarly, Figure 8 illustrates the performance for Scenario (2), while Figure 9 shows the results for the Mondego River in Scenario (1).

–
**Tejo River Results**
For the Tejo River, the model accurately captures short-term trends, especially for today’s flow, with acceptable accuracy. However, as predictions extend further into the future, slight increases in RMSE reflect growing uncertainty. The model effectively tracks peaks and troughs but shows some under- and overpredictions as the forecast horizon grows. Both scenarios demonstrate reliable short-term forecasting, and Scenario (2) incorporates rainfall data to provide an alternative view of the influence of precipitation.–
**Mondego River Results**
In contrast, the Mondego River models exhibit lower RMSE values, with 49.05 m^3^/s for 1-day forecasts (Scenario (1)). This indicates a higher predictive accuracy compared to the Tejo River. Similar trends of increasing RMSE are observed with longer forecast horizons, although the absolute errors remain smaller. The bias values are closer to zero, suggesting more balanced predictions.

## 6. Conclusions and Future Work

This study explored the rich datasets provided by a comprehensive multi-sensor river network to develop MLP-based forecasting models for predicting daily river discharges under various hydraulic conditions. The work was demonstrated for two major Portuguese basins, and focused on very high river flow periods. The selection of MLPs was based on a comparative analysis of deep learning methods applied to the most complex of the two use cases. Then, several models with MLPs were built to achieve adequate prediction accuracy. Although MLPs adequately record general discharge patterns, forecasting peak flows from intense rainfall or abrupt dam releases remains difficult. These infrequent and rapidly occurring phenomena bring about ambiguities that complicate accurate predictions. However, the developed models continue to provide significant early warnings, making them an invaluable asset for proactive flood control. In addition, they compare favorably with traditional persistence approaches based on the last measured river flow.

The general findings for the Tejo and Mondego rivers are both satisfactory and justifiable. Despite increasing prediction errors during high discharge events, MLP models consistently provide prompt notifications for elevated water levels. They surpass or equal other ML methodologies under standard flow circumstances, demonstrating their dependability. In addition, the models accommodate various inputs, including historical flows, precipitation data, and dam operations, showing adaptability to the different circumstances in the two basins.

Enhancing peak flow predictions requires training techniques centered on flood occurrences, such as oversampling peak flow data or assigning more weight to mistakes in high-discharge scenarios. Furthermore, the use of near-real-time precipitation data and detailed dam operational information should facilitate the detection of rapid alterations often seen before floods.

Subsequently, we want to evaluate our models over a broader spectrum of flood scenarios to ensure the inclusion of diverse meteorological and hydrological trends. Collaboration with water management authorities and civil protection organizations is essential to synchronize the results of the models with daily activities in flood-prone regions. Focusing on these peak flow occurrences and integrating more comprehensive information, our goal is to improve MLP-based forecasting into a reliable decision support instrument for severe flood events.

Although the MLP model demonstrated consistent performance across multiple scenarios, we emphasize that these findings are specific to the particular datasets, hydrological conditions, and hyperparameter configurations investigated in this study. Specifically, we evaluated MLPs alongside other conventional ML approaches (SVM, RF, ELM), with the MLP approach generally outperforming these methods and achieving lower RMSE values. However, we do not claim that MLPs generally outperform all deep learning architectures, such as transformer-based models, CNN-LSTM hybrids, or GRU variants. The relative advantage we observed for MLPs in our experiments may be related to the size of the dataset, the nature of flows controlled by dams, and the complexity of model tuning. Also, in the subject of model tuning, future studies will consider dynamic hyperparameter tuning—e.g., Bayesian Optimization and meta-learning—to help models be more flexible in response to seasonal and environmental fluctuations.

Future work could include testing more advanced deep learning techniques—for example, GRU models, transformer networks, or attention-based architectures—to assess whether more advanced or specialized approaches might enhance predictions, particularly for extreme flow events. In this regard, our study should be considered a systematic but non-exhaustive comparison, demonstrating that MLPs are a practical and reliable choice under a wide range of conditions rather than definitively claiming their dominance in all areas of hydrological forecasting.

## Figures and Tables

**Figure 1 sensors-25-02154-f001:**
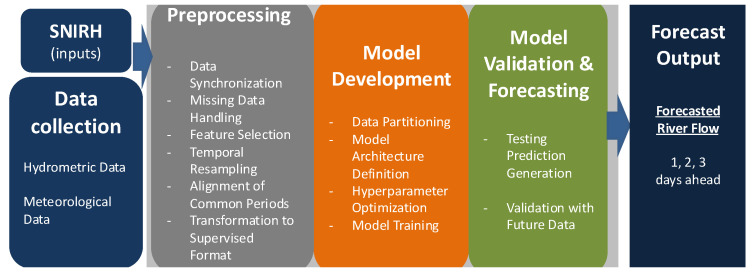
Methodology architecture diagram.

**Figure 2 sensors-25-02154-f002:**
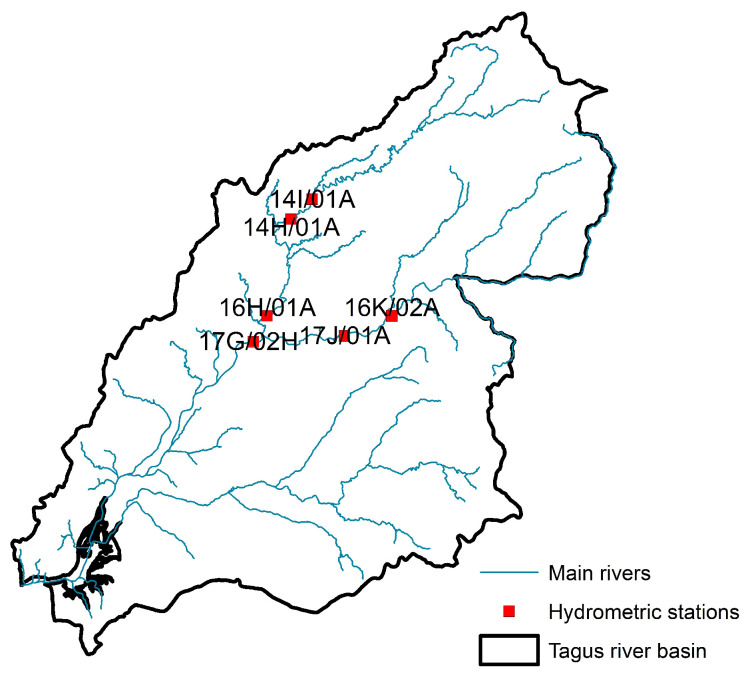
Locations of hydrometric measuring stations in the Tejo.

**Figure 3 sensors-25-02154-f003:**
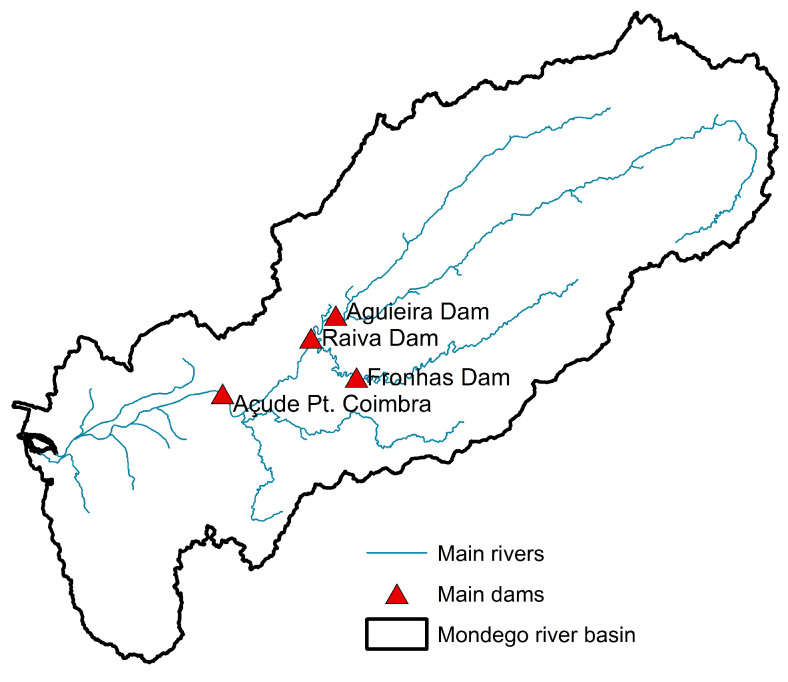
Locations of hydrometric measuring stations in the Mondego basins.

**Figure 4 sensors-25-02154-f004:**
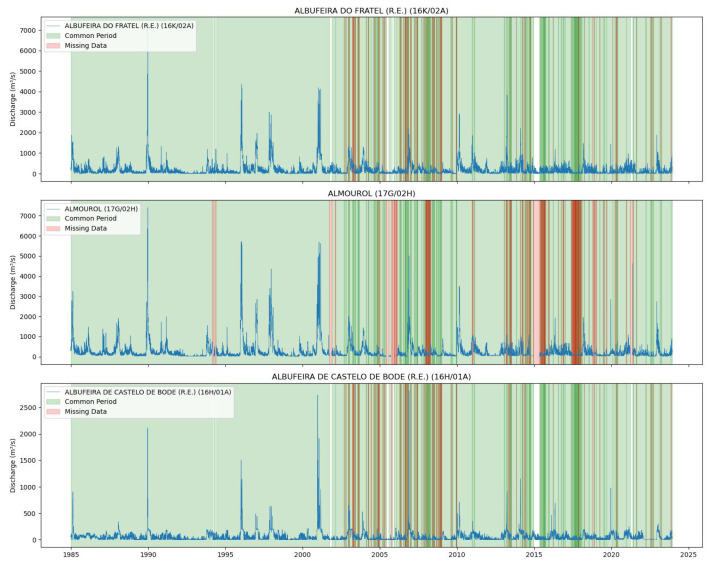
Common data periods at measuring stations for the Tejo River (Scenario (1)).

**Figure 5 sensors-25-02154-f005:**
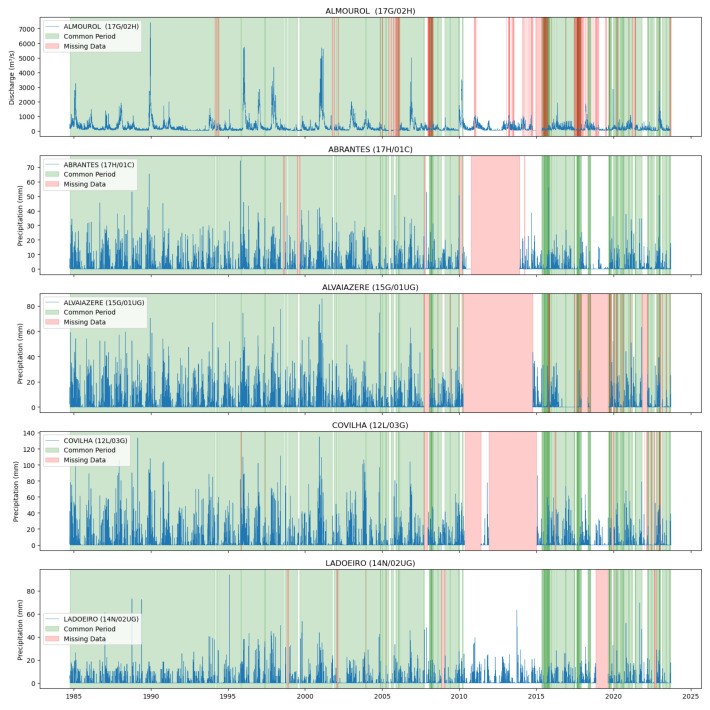
Common data periods at measuring stations for the Tejo River (Scenario (2)).

**Figure 6 sensors-25-02154-f006:**
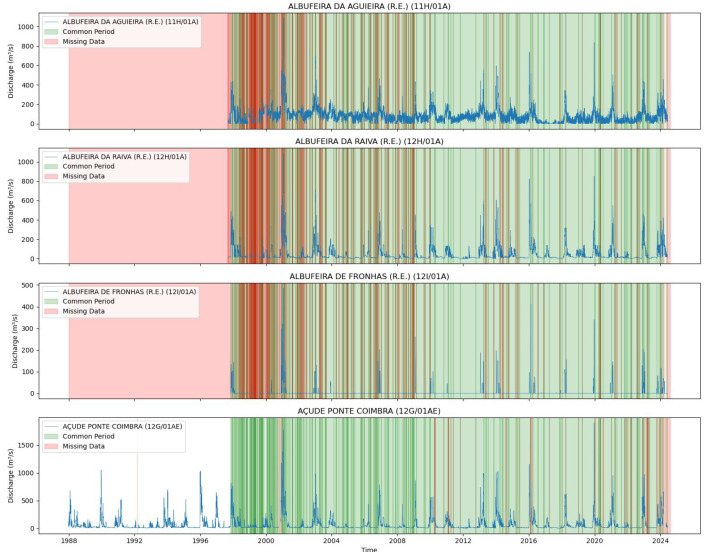
Common data periods at measuring stations for the Mondego River (Scenario (1)).

**Figure 7 sensors-25-02154-f007:**
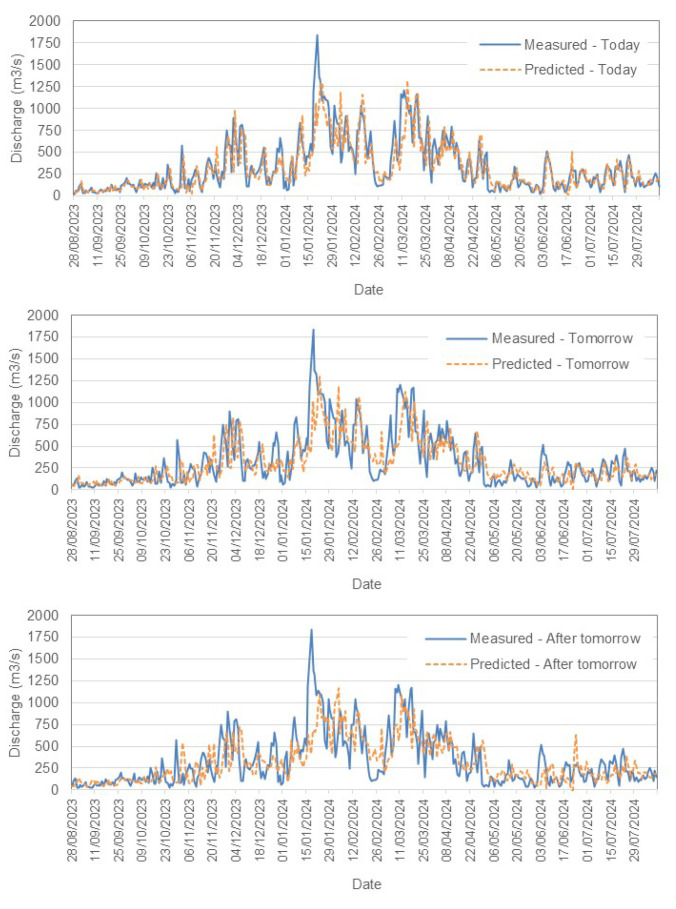
Comparison between measured and predicted daily discharge (m^3^/s) in the Tejo River, Scenario (1).

**Figure 8 sensors-25-02154-f008:**
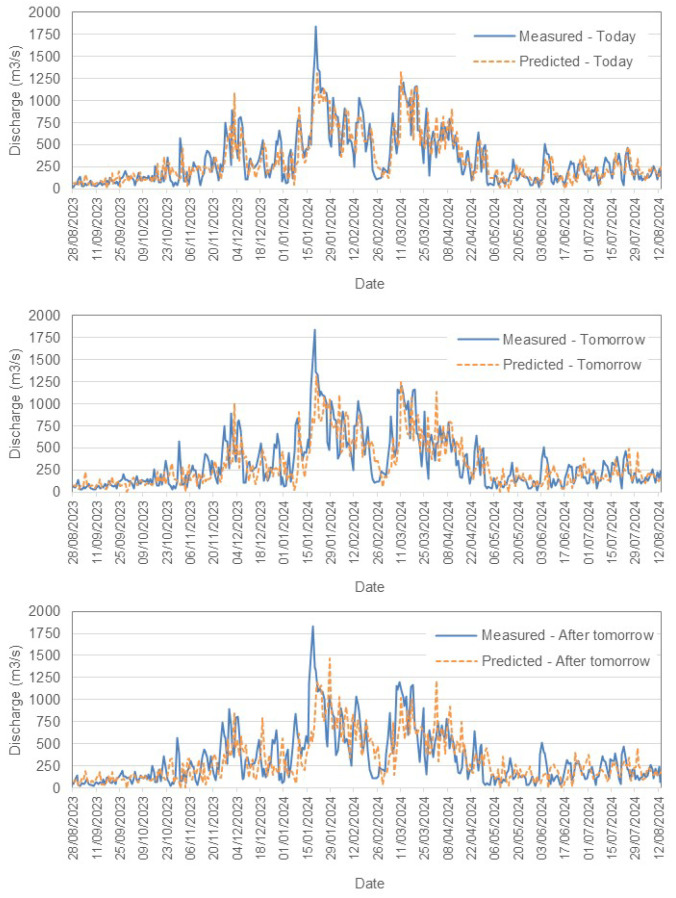
Comparison between measured and predicted daily discharge (m^3^/s) in the Tejo River, Scenario (2).

**Figure 9 sensors-25-02154-f009:**
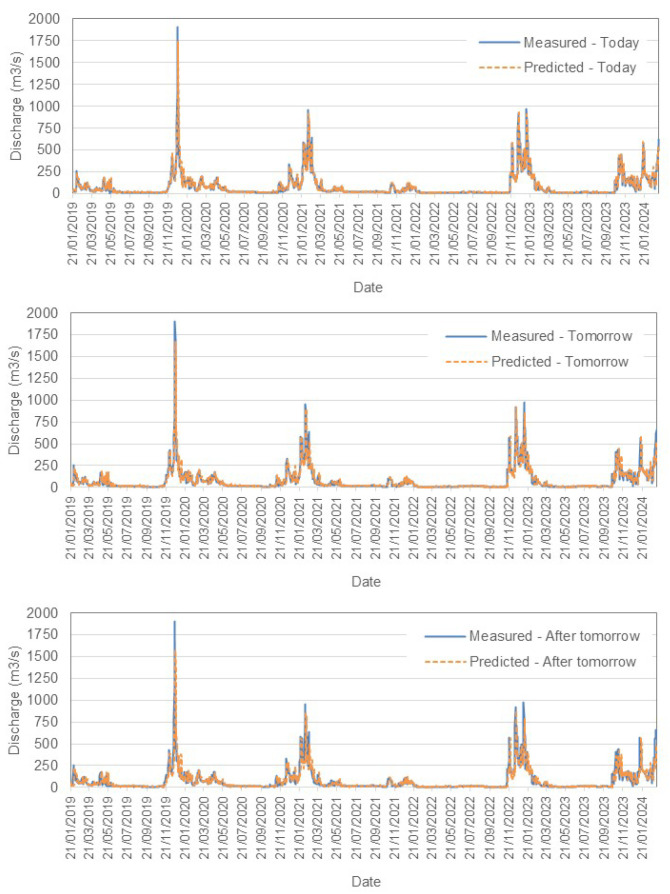
Comparison between measured and predicted daily discharge (m^3^/s) in the Mondego River, Scenario (1).

**Table 1 sensors-25-02154-t001:** LSTM model configurations for use scenarios.

Sc	Cfg.	LSTM Layers (Units, Act., Seq.)	Dropout	Output (Units, Act.), Optimizer, Batch, Epochs
a	1	96, ReLU, T → 96, ReLU, F → 96, ReLU, F	0.3/-/0.2	2, Softplus, Adam (lr = 0.00929), 32, 50
b	1	128, ReLU, T → 128, ReLU, F → 96, ReLU, F	0.4/0.3/-	2, Linear, Adam (lr = 0.00691), 32, 50
	2	50, ReLU, F	-	2, Linear, Adam, 32, 100
c	1	64, ReLU, T → 128, ReLU, F → 40, ReLU, F	0.1/-/0.1	2, Linear, Adam (lr = 0.00576), 32, 50
d	1	32, LSTM, T → 90, LSTM, F → 96, LSTM, F	0.1/-/-	1, Linear, Adam (lr = 0.00022), 32, 50
	2	50, ReLU, T → 30, ReLU, F	0.1	1, SELU, Adam, 16, 1000

**Table 2 sensors-25-02154-t002:** MLP configurations for use scenarios.

Sc	Cfg.	Hidden Layers (Units, Act.)	L2 Reg	Output (Units, Act.), Optimizer, Batch, Epochs
a	1	[90, 90], ReLU	0.01	2, SELU, Adam (lr = 0.001), 16, 100
b	1	[150, 40], ReLU	0.01	2, SELU, Adam (lr = 0.001), 16, 13
	2	[150, 40], ReLU	0.01	2, SELU, Adam (lr = 0.001), 16, 300
c	1	[90, 90], ReLU	0.01	2, SELU, Adam (lr = 0.001), 16, 100
	2	64, ReLU → 90, ReLU	0.01	2, ReLU, Adam (lr = 0.0003916), 16, 100
d	1	150, ReLU → 150, ReLU → 90, ReLU	0.01	1, ReLU, Adam (lr = 0.0003097), 16, 100

**Table 3 sensors-25-02154-t003:** ELM configurations for use scenarios.

Sc	Cfg.	Hidden Neurons	Activation
a	1	90	sigm
	2	150	sigm
b	1	200	tanh
c	1	50	tanh
d	1	90	sigm

**Table 4 sensors-25-02154-t004:** SVM configurations for use scenarios.

Sc	Cfg.	C	Gamma	Epsilon
a	1	3	scale	0.02
	2	5	0.001	0.02
b	1	10	scale	0.02
c	1	100	scale	0.2
	2	100	scale	0.2
d	1	100	scale	0.5

**Table 5 sensors-25-02154-t005:** RF model configurations for scenarios.

Sc	Cfg.	n_Estimators, max_Depth, min_Samples_Split
a	1	100, Default (None), Default (2)
b	1	100, Default (None), Default (2)
c	1	100, None (Day-1) → 10 (Day-2), 10
d	1	100, 30, 10

**Table 6 sensors-25-02154-t006:** Comparative performance (RMSE in m^3^/s) of ML and DL models for river flow forecasting. Note: For scenarios a, b, and c (2-day forecasts), each cell shows (RMSE for Day 1, RMSE for Day 2). For scenario d (1-day forecast), only a single RMSE value is shown. Validation periods differ by scenario. Bold values indicate the best (lowest) RMSE for MLP among the compared models for the corresponding forecast horizon.

Sc.	Validation Period	Input/Output	Days	MLP	LSTM	SVM	RF	ELM
				162.65	168.70,	367.23,	155.71,	326.42,
a	2022–08 to 2023–09	[0,2] → [4]	2	**227.02**	239.71	367.31	264.51	334.82
					203.01,		307.51,	
					231.81		308.27	
				152.71,	141.85,	198.12,	148.79,	218.08,
b	2022–08 to 2023–09	[0,2,3] → [4]	2	**213.88**	216.48	230.45	261.46	255.18
				149.14,				
				227.87				
				136.71,	136.02,	153.17,	149.53,	103.82,
c	2022–08 to 2023–09	[0,1,2,3,4] → [4]	2	206.49	212.92	200.09	257.72	**179.47**
				141.39,				
				217.94				
d	2003–03 to 2004–11	[0,2,4] → [4]	1	**104.53**	121.51	105.75	119.11	223.36
					356.93			

**Table 7 sensors-25-02154-t007:** Selected features for the Tejo River.

Scenario	Station/Features
(1) Inputs	Fratel (R.E.) (16K/02A)/Average daily dam outflow discharge (m^3^/s)
	Almourol (17G/02H)/Average daily river discharge (m^3^/s)
	Castelo de Bode (R.E.) (16H/01A)/Average daily dam outflow discharge (m^3^/s)
(1) Output	Almourol (17G/02H)/Average daily river discharge (m^3^/s)
(2) Inputs	Abrantes (17H/01C)/Daily precipitation(mm)
	Alvaiázere (15G/01UG)/Daily precipitation (mm)
	Covilhã (12L/03G)/Daily precipitation (mm)
	Ladoeiro (14N/02UG)/Daily precipitation (mm)
	Almourol (17G/02H)/Average daily river discharge (m^3^/s)
(2) Output	Almourol (17G/02H)/Average daily river discharge (m^3^/s)

**Table 8 sensors-25-02154-t008:** Selected features for the Mondego River.

Scenario	Station/Features
(1) Inputs	Albufeira da Aguieira (R.E.) (11H/01A)/Average daily dam outflow discharge (m^3^/s)
	Albufeira da Raiva (R.E.) (12H/01A)/Average daily dam outflow discharge (m^3^/s)
	Albufeira de Fronhas (R.E.) (12I/01A)/Average daily dam outflow discharge (m^3^/s)
	Açude Ponte Coimbra (12G/01AE)/Average daily weir outflow discharge (m^3^/s)
(1) Output	Açude Ponte Coimbra (12G/01AE)/Average daily weir outflow discharge (m^3^/s)

**Table 9 sensors-25-02154-t009:** Training, Testing, and Validation Sets for the Tejo and Mondego Rivers.

River	Scenario	Training Set	Testing Set	Validation Set
Tejo	(1)	8042 samples, 60 features	2011 samples, 60 features	350 samples, 60 features
	(2)	7120 samples, 100 features	1780 samples, 100 features	350 samples, 100 features
Mondego	(1)	3643 samples, 80 features	911 samples, 80 features	1873 samples, 80 features

**Table 10 sensors-25-02154-t010:** MLP configurations for scenario in the Mondego River.

Sc.	Cfg.	Hidden Layers (Units, Act.)	L2 Reg	Output (Units, Act.), Optimizer, Batch, Epochs
(1)	MLP1	[50], ReLU	0.1	3, linear, Adam, 32, 50
(1)	MLP2	[90, 90], ReLU	0.1	3, linear, Adam, 32, 100
(1)	MLP3	[150, 40], ReLU	0.1	3, linear, Adam, 32, 300

**Table 11 sensors-25-02154-t011:** Comparative RMSE (m^3^/s) of different MLP models for Scenario (1) in the Mondego River. Note: Bold values indicate the lowest RMSE (best performance) for each forecast horizon.

Model	RMSE (m^3^/s) Today	RMSE (m^3^/s) Tomorrow	RMSE (m^3^/s) Day After	Validation Period
MLP1	26.79	**27.91**	**28.24**	2024-01-01 to 2024-08-12
MLP2	24.51	44.87	66.21	2024-01-01 to 2024-08-12
MLP3	34.01	45.54	54.04	2024-01-01 to 2024-08-12

**Table 12 sensors-25-02154-t012:** Evaluation of MLP model performance for Tejo and Mondego rivers.

River	Scenario	Forecast Horizon	RMSE (m^3^/s)	Bias (m^3^/s)
Tejo	(1)	1-Day (Today)	163.1	−22.5
	(1)	2-Day (Tomorrow)	212.1	−20.0
	(1)	3-Day (After Tomorrow)	228.4	−18.9
	(2)	1-Day (Today)	169.9	−5.6
	(2)	2-Day (Tomorrow)	215.1	−25.8
	(2)	3-Day (After Tomorrow)	232.6	−21.5
Mondego	(1)	1-Day (Today)	49.05	2.08
	(1)	2-Day (Tomorrow)	72.80	−0.76
	(1)	3-Day (After Tomorrow)	86.50	−3.62

## Data Availability

The dataset used in this study originates from real-time sensor network observations provided by the Portuguese National Water Resources Information System (SNIRH). This network comprises hydrological and meteorological sensors that continuously monitor variables such as river discharge, precipitation, and water levels across multiple gauging stations. These sensor-based observations ensure high temporal resolution and reliability of the data, which is critical for developing and validating machine learning-based hydrological forecasting models. The dataset is publicly accessible and can be retrieved from the SNIRH portal at: https://snirh.apambiente.pt/ (accessed on 30 December 2024).

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
