# Peer review of "Deep Learning-Based River Flow Forecasting with MLPs: Comparative Exploratory Analysis Applied to the Tejo and the Mondego Rivers"

_sensors, 2025, doi:10.3390/s25072154_

Round 1
Reviewer 1 Report
Comments and Suggestions for Authors
This paper proposes the use of a Multilayer Perceptron (MLP) to forecast the flow of two rivers in Portugal. The workflow is described in detail, and the performance of the MLP is compared with other machine learning (ML) and deep learning (DL) methods. However, I have several concerns about this study:
1. The application of MLP for river flow forecasting is not novel. Similar studies have been conducted more than two decades ago, such as the work https://www.sciencedirect.com/science/article/abs/pii/S146419090185005X. Although the authors compare MLP with other methods, such as Random Forest (RF) and Long Short-Term Memory (LSTM), the comparison is not systematic. I recommend conducting a more thorough and detailed comparison across different approaches. Additionally, the innovation and contribution of this study should be clarified and emphasized.
2. The results show that LSTM performs worse than MLP in this study, which is surprising. LSTM is widely regarded as being superior to MLP for time-series data due to its ability to capture temporal dependencies. The authors should ensure the LSTM model is properly configured, as an improperly tuned LSTM could explain its underperformance.
3. The study lacks critical details about the ML and DL models. For example, how many layers were used in the MLP, and what was the size of the hidden layers? It is also suggested to briefly introduce the theoretical background of these methods. No equations are provided.
4. The writing of the paper needs improvement. For example:
Abbreviations should be introduced when they first appear, and the full name should not be repeated later.
Excessive use of bold and underlined text for emphasis should be avoided.
I suggest using high-quality papers as templates to improve the writing style, such as
https://hess.copernicus.org/articles/26/5449/2022/hess-26-5449-2022.pdf https://www.sciencedirect.com/science/article/abs/pii/S136481522200281X
5. In Table 4, the absolute value of the bias is much smaller than RMSE. The method for calculating bias is unclear. The authors should provide equations for both bias and RMSE to enhance transparency and reproducibility.
6. Figures 1 and 2 can be combined into a single figure with two subfigures for better clarity and conciseness.
7. In Lines 150–159, the description of PSO and GA as forest models is inaccurate, as these are optimization methods. The authors are encouraged to explore and introduce more advanced DL techniques, such as transformer-based and attention-based approaches. Examples include recent studies such as:
https://www.sciencedirect.com/science/article/pii/S0022169424013192 https://link.springer.com/article/10.1007/s40808-024-02088-y
8. The paper sometimes incorrectly groups MLP, RF, and SVM under deep learning. These methods are typically categorized as machine learning, not deep learning. The authors should correctly differentiate between ML and DL methods.
Reviewer 2 Report
Comments and Suggestions for Authors
The manuscript explores one of the earliest practical ANNs, Multilayer Perceptrons (MLPs), for forecasting river flow. While using a classic method in solving a well-studied phenomenon could still offer some value, the manuscript suffers various flaws, raising serious concerns. If not rejected, significant improvements are required to be considered for the review process.
The manuscript suffers numerous issues. Here, only some examples are provided:
The introduction lacks a clear research question and hypothesis formulation. It also does not explicitly state how it advances the field of hydrological forecasting.
The related work section is superficial. The authors list various models but fail to critically compare their strengths and weaknesses in time-series forecasting.
Methodology must be significantly improvemed before considering the manuscript for the peer-review process. The authors must provide a reasonable introduction to the tools and methods. Using practical technical language and effective visualizations provided in research works with high pedagogical values including but not limited to https://doi.org/10.1007/978-3-031-34593-7_69 could help to address this serious concern.
The formatting and the way that the materials are represented need significant improvements. For instance, using bold formatting in the text does not seem common (e.g., in the Introduction section). Similarly, for underlines in section 2 (Related works).
The literature review must be significantly improved. A shallow revision on various available methods for the same problem does not compensate a comprehensive investigation for providing a reasonable understanding of the recent similar works using the same method.
Most cited works are older than 2020, missing recent advancements in AI-based hydrological forecasting. The paper repeatedly cites previous works by the authors, potentially inflating their contribution.
Before considering the manuscript for review process, the authors must address this serious concern by exploring related works similar to DOI: 10.1007/978-3-031-34593-7_69. In this example, just in the introduction section over 61 relevant studies were explored and a reasonable litreture review is provided.
References must be checked for inaccuracies and possible mistakes. e.g. the XXXX–XXXX does not seem to be correct in "Khan, M.T.; Shoaib, M.; Hammad, M.; Salahudin, H.; Ahmad, F.; Ahmad, S. Application of674 Machine Learning Techniques in Rainfall–Runoff Modelling of the Soan River Basin, Pakistan.675 Water 2021, 13, XXXX–XXXX."
The study raises serious concerns about reproducibility. The authors fail to mention whether their dataset is publicly available for reproducibility. Also, the Lack of precise model training details raises serious concerns about model validity. Batch sizes, learning rates, weight initialization techniques, and optimizer settings are not clearly stated. The lack of code availability makes it impossible to replicate their results.
The authors may overclaim the effectiveness of their method without sufficient empirical evidence. They state that MLPs perform best but do not test alternative deep learning models.
The manuscript has serious writing, structure, and clarity Issues. It lacks a coherent structure. For instance, in the abstract, The phrase "attain reasonable accuracy" is vague and unscientific.
This sentence is too long, convoluted, and redundant. It should be rewritten concisely: "These models strive to deliver precise and timely predictions to support water resource management and decision-making processes by leveraging innovative data-driven methodologies combined with existing hydrological and meteorological data sources."
Getting help from a native English speaker for a thorough revision is seriously recommended.
Round 2
Reviewer 1 Report
Comments and Suggestions for Authors
The authors have well addressed my previous comments.
Author Response
We thank the reviewer for the positive feedback and are pleased that the manuscript was up to standards.